Molecular characterization, expression patterns and cellular localization of BCAS2 gene in male Hezuo pig

Tang Yuran 1
Zhang Bo 1
Shi Haixia 1
http://orcid.org/0000-0001-8701-0647 Yan Zunqiang 1
Wang Pengfei 1
Yang Qiaoli 1
http://orcid.org/0000-0001-9467-3406 Huang Xiaoyu 1
Gun Shuangbao 1 2 gunsbao056@126.com
1 College of Animal Science and Technology, Gansu Agricultural University , Lanzhou, Gansu , China
2 Gansu Research Center for Swine Production Engineering and Technology , Lanzhou, Gansu , China
Oliveira Sonia
Electronic publication date: 2023 Oct 24
Publication date: 2023
Volume: 11
Electronic Location ID: e16341
Received 2023 Mar 31; Accepted 2023 Oct 3
Copyright: © 2023 Tang et al.
Copyright year: 2023
Copyright holder: Tang et al.
License: This is an open access article distributed under the terms of the Creative Commons Attribution License, which permits unrestricted use, distribution, reproduction and adaptation in any medium and for any purpose provided that it is properly attributed. For attribution, the original author(s), title, publication source (PeerJ) and either DOI or URL of the article must be cited.
License URL: https://creativecommons.org/licenses/by/4.0/

Keywords: Hezuo pig, BCAS2, Molecular characterization, Testicular development, Spermatogenesis

Funding: Gansu Local Pig Germplasm Resources GSLK-2021-13 This work was supported by the Protection and Quality Improvement of Gansu Local Pig Germplasm Resources (No. GSLK-2021-13). The funders had no role in study design, data collection and analysis, decision to publish, or preparation of the manuscript.

==============================
Background

Breast carcinoma amplified sequence 2 (BCAS2) participates in pre-mRNA splicing and DNA damage response, which is implicated in spermatogenesis and meiosis initiation in mouse. Nevertheless, the physiological roles of BCAS2 in the testes of large mammals especially boars remain largely unknown.

Methods

In this study, testes were collected from Hezuo pig at three development stages including 30 days old (30 d), 120 days old (120 d), and 240 days old (240 d). BCAS2 CDS region was firstly cloned using RT-PCR method, and its molecular characteristics were identified using relevant bioinformatics software. Additionally, the expression patterns and cellular localization of BCAS2 were analyzed by quantitative real-time PCR (qRT-PCR), Western blot, immunohistochemistry and immunofluorescence.

Results

The cloning and sequence analysis indicated that the Hezuo pig BCAS2 CDS fragment encompassed 678 bp open reading frame (ORF) capable of encoding 225 amino acid residues, and possessed high identities with some other mammals. The results of qRT-PCR and Western blot displayed that BCAS2 levels both mRNA and protein were age-dependent increased (p < 0.01). Additionally, immunohistochemistry and immunofluorescence results revealed that BCAS2 protein was mainly observed in nucleus of gonocytes at 30 d testes as well as nucleus of spermatogonia and Sertoli cells at 120 and 240 d testes. Accordingly, we conclude that BCAS2 is critical for testicular development and spermatogenesis of Hezuo pig, perhaps by regulating proliferation or differentiation of gonocytes, pre-mRNA splicing of spermatogonia and functional maintenance of Sertoli cells, but specific mechanism still requires be further investigated.

Introduction

Breast carcinoma amplified sequence 2 (BCAS2) is a novel gene proposed by differential display (DD) technique in human breast cancer cell line (Nagasaki et al., 1999; Maass et al., 2002). It was originally proposed as DNA amplified in mammary carcinoma (DAM1) (Nagasaki et al., 1999), then named BCAS2. BCAS2 as negative regulator of tumor suppressor p53 through direct interaction affects cell viability (Kuo et al., 2009), and the expression of BCAS2 has potentially correlation with the aggressive breast cancer (Qi et al., 2005; Salmerón-Hernández et al., 2019), prostate cancer (Kuo et al., 2015) and esophageal cancer (Lang & Zhao, 2018). BCAS2, as a core component of the cell division cycle 5-like/Pre-mRNA-processing factor 19 (CDC5L/Prp19) complex (Ajuh et al., 2000; Chan et al., 2003), is involved in pre-mRNA splicing (Krämer, 1996), the machinery that introns are removed from mRNA precursors (Black, 2003). Previous reports have confirmed that the depletion of BCAS2 in both yeast (Chen et al., 1998, 1999) and Drosophila (Chen et al., 2013; Chou et al., 2015) resulted in the accumulation of pre-mRNA. As an important component of CDC5L/Prp19 complex, BCAS2 also has been reported to play critical role in DNA damage response through interacting directly with replication protein A complex (RPA complex) (Xu et al., 2015; Wan & Huang, 2014). BCAS2 also takes part in spindle assembly and chromosome arrangement (Hofmann et al., 2013), which is essential for mitotic initiation (Boudrez et al., 2000). Furthermore, BCAS2 can enhance the transcriptional activity and protein stability of androgen receptor (AR) (Kuo et al., 2015) and estrogen receptor (ER) (Qi et al., 2005; Salmerón-Hernández et al., 2019), affect the survival of hematopoietic stem and progenitor cells (HSPCs) during hematopoiesis development (Yu et al., 2019), interfere lens development and differentiation (Liao et al., 2015), and regulate neuronal dendrite growth and morphology (Huang et al., 2016). Importantly, BCAS2 is implicated in the protection of genomic integrity during preimplantation development, and oocyte-specific knockout has been shown to cause the developmental arrest of embryos with defects in DNA replication and accumulation in DNA damage (Xu et al., 2015). Recently, studies have already demonstrated that BCAS2 can modulate alternative splicing during spermatogenesis in mice, and specific deletion of this gene can lead to abnormal initiation of spermatogonia meiosis and loss of fertility (Liu et al., 2017).

Despite its importance, previous studies on BCAS2 gene function mainly focused on the regulation of cancer malignancy including breast cancer (Qi et al., 2005; Salmerón-Hernández et al., 2019), prostate cancer (Kuo et al., 2015) and esophageal cancer (Lang & Zhao, 2018), and there are largely unknown about its physiological roles in normal cells, particularly in germ cells. Existing studies have demonstrated that BCAS2 has great influence on early embryonic development and reproductive ability in mice (Xu et al., 2015; Liu et al., 2017). However, there are few reports regarding BCAS2 in male domestic animals, especially boars. Accordingly, it is extremely necessary to elucidate the potential roles of BCAS2 gene during spermatogenesis in porcine testis. Spermatogenesis in mammals is a highly intricate cascade process finely controlled by multiple genes whose expression has been investigated at both transcriptional and translational levels (Gan et al., 2013; Chalmel & Rolland, 2015), which is smoothly carried out to directly determine semen quality and influence fertility for male livestock animals (Zhao et al., 2019; La & Hobbs, 2019). Additionally, the study of concrete regulation mechanisms of genes correlated with spermatogenesis can further accurately reflect the reproductive regularity of economic animals, which can provide new insights for improving the fecundity of domestic animals (Fischer et al., 2015; Du et al., 2014; Ran et al., 2017).

Hezuo pig is a miniature plateau-type primitive indigenous breed in China, which has the characteristics of early sexual maturity and low fertility simultaneously (Ma et al., 2019), so it is a good choice for the study of mammalian fecundity. In this work, the testes of Hezuo pig were collected from different developmental phases in order to obtain the full-length coding sequence (CDS) of BCAS2, identify its molecular characterization, examine its expression patterns at the mRNA and protein levels, as well as explore the localization of BCAS2-positive cells. This is of great significance to preliminarily understand the biological functions and regulatory mechanisms of BCAS2 during testicular development and spermatogenesis in boars, and even in other male mammals. In addition, these results will also provide additional clues for further understanding of reproductive related diseases or cancers in animals and even humans.

Materials and Methods

Experimental animals and design

A total of nine healthy male Hezuo pig, derived from three different reproductive stages including 30 days old (30 d; n = 3), 120 days old (120 d; n = 3), and 240 days old (240 d; n = 3), were obtained from Hezuo, Gannan Tibetan Autonomous Prefecture, Gansu Province, China. These Hezuo pigs were raised under the same environmental conditions accompanied by natural light and free access to food as well as water. Testes at three reproductive stages were collected, one sample was kept at −80 °C until the extraction of total RNA and protein, and the other part was fixed in 4% paraformaldehyde (Solarbio, Beijing, China) to be used for making paraffin sections. Besides, after slaughtering under barbiturate anesthesia, samples including heart, liver, spleen, lung and kidney were promptly collected from 240 d Hezuo pigs, stored at −80 °C for RNA extraction. All animals in the study were strictly managed in line with animal care and experimental guidelines, which was subject to approval by the Animal Care Committee of Gansu Agricultural University (Approval No. GAU-LC-2019-154).

Total RNA extraction and cDNA synthesis

Total RNA from Hezuo pig tissue samples was extracted by operation instructions of TRIzol Reagent (TransGen Biotech, Beijing, China). The purity and concentration of extracted total RNA were examined utilizing NanoDrop ND-2000 spectrophotometer (Thermo Fisher Scientific, Niederelbert, Germany), and its integrity was evaluated with 1% agarose gel electrophoresis. Each RNA sample with equal concentration (500 ng) was reverse transcribed into cDNA by using PrimeScript™ RT Reagent Kit with gDNA Eraser (TaKaRa, Dalian, China), and then preserved at −20 °C for further use.

Full-length cDNA cloning of BCAS2 gene

Primer pairs were designed with Primer-BLAST in NCBI according to sequence information of Sus scrofa BCAS2 gene retrieved from GenBank with Accession No. NM_001244424.1. Primers were synthesized commercially by Genewiz Co. Ltd. (Suzhou, China). The details of primer pairs are summarized in Table 1. The CDS region sequence of BCAS2 gene was amplified by utilizing cDNA of testis samples. The PCR reaction was carried out in 20 µL final volume containing 10 µL of 2×Taq PCR Master Mix, 6 µL of RNase free ddH2O, 2 µL of cDNA, and 1 µL of each forward and reverse primer. The PCR procedure was as follows: one cycle of 95 °C for 5 min; then 30 cycles of 94 °C for 30 s, 58 °C for 30 s, and 72 °C for 1 min; and finally one cycle of 72 °C for 10 min. Amplified cDNA products were separated using 1% agarose gel and purified according to TIANgel Midi Purification Kit (TransGen Biotech, Beijing, China). Subsequently, purified PCR products were inserted into the pMD™19-T vector following pMD™19-T Vector Cloning kit (TaKaRa, Dalian, China), and transformed into E.coli DH5α competent cells. Five independent positive clones were arbitrarily selected and commercially sequenced by the Genewiz Co. Ltd. (Suzhou, China).

Table 1 List of the primers used in this study.

Gene	Accession no.	Primer sequence (5′-3′)	Size (bp)	Purpose	
BCAS2	NM_001244424.1	F: TCAGGCGAGCCCAAGAATG	730	cDNA cloning	
R: GAAAGCCTTCCTTCCTACCCG	
BCAS2	NM_001244424.1	F: CAGCCCCAGATTATTCCGCT	114	qRT-PCR	
R: GGGCCGGAAGTTCATATCGT	
β-actin	XM_021086047.1	F: GATGACGATATTGCTGCGCTC	212	qRT-PCR	
R: TCGATGGGGTACTTGAGGGT	
Note:

F, forward primer; R, reverse primer.

Bioinformatics analysis

Multiple data banks and online software were used to shed light on the molecular characterization of Hezuo pig BCAS2 gene acquired by cloning. The entire open reading frame (ORF) sequence of BCAS2 gene was identified by ORF Finder (https://www.ncbi.nlm.nih.gov/orffinder/). The conserved domains of BCAS2 protein were searched by Conserved Domain Database (CCD) in NCBI server (https://www.ncbi.nlm.nih.gov/Structure/cdd/wrpsb.cgi). The BCAS2 CDS sequences obtained from Hezuo pig testes were compared with those present in NCBI using DNAMAN software. The basic physicochemical properties of the deduced BCAS2 protein were inferred by ProtParam Server (https://web.expasy.org/protparam/). The phosphorylated sites were analyzed using NetPhos 3.1 Server (https://services.healthtech.dtu.dk/services/NetPhos-3.1/). The subcellular localization of BCAS2 protein was speculated by using PSORTII Prediction server (https://psort.hgc.jp/form2.html). The hydrophobicity was determined by Protscale (http://web.expasy.org/ProtScale/). The signal peptide cleavage sites were predicted by the SignalP 4.1 Server (https://services.healthtech.dtu.dk/service.php?SignalP-4.1). Transmembrane domains were searched by TMHMM-2.0 Server (https://services.healthtech.dtu.dk/service.php?TMHMM-2.0). The secondary and tertiary structures of BCAS2 protein were predicted by using SOPMA (https://npsa-prabi.ibcp.fr/cgi-bin/npsa_automat.pl?page=npsa_sopma.html) and SWISS-MODEL (https://swissmodel.expasy.org/), respectively. Multiple sequence alignments from different species were analyzed using BLAST algorithm in NCBI server (http://blast.ncbi.nlm.nih.gov/Blast.cgi). The phylogenetic tree based on the nucleotide sequences was constructed by using the neighbor-joining (NJ) method in MEGA 7.0 software, and the reliability level of each branch was tested by 1,000 bootstrap replicates.

Quantitative real-time polymerase chain reaction (qRT-PCR)

The primer pairs used for qRT-PCR were designed and synthesized according to the sequence information of BCAS2 and β-actin from NCBI, and the details are shown in Table 1. The β-actin was used as a housekeeping gene for the normalization of BCAS2 mRNA expression. The qRT-PCR was carried out in LightCycler 96 Real-Time System (Roche, Basel, Switzerland) following optimized procedures: one cycle of 95 °C for 3 min; 40 cycles of 95 °C for 15 s, 60 °C for 15 s, and 72 °C for 20 s. The amplification reaction system (20 µL) comprised of 0.8 µL of each forward and reverse primer (10 µM), 2 µL of cDNA template (100 ng), 6.4 µL of RNase free ddH2O, and 10 µL of 2× SYBR Premix Ex Taq (Takara, Dalian, China). The relative expression level of BCAS2 mRNA was evaluated relative to that of β-actin using the 2−∆∆Ct method.

Western blot

Total proteins of testes at different development stages were extracted by the protein extraction kit (Solarbio, Beijing, China) containing radioimmunoprecipitation assay (RIPA) and phenylmethanesulfonyl fluoride (PMSF) according to the kit manual. Protein concentrations were examined utilizing commercial BCA protein assay kit (Beyotime, Shanghai, China). The protein lysates were mixed using 4× SDS-PAGE loading buffer (Solarbio, Beijing, China), and then denatured by boiling at 95 °C for 10 min. The denatured protein with equal concentrations (20 µg) were separated by 12% SDS-PAGE, and then electrotransferred onto 0.45 μm polyvinylidene difluoride (PVDF) blotting membranes (Millpore, Bedford, MA, USA) using a wet transfer system for 90 min at 100 V. The blotting membranes were blocked by using 5% skimmed milk, and incubated with either rabbit anti-BCAS2 polyclonal antibody (1:1,000 dilution; Catalog No. DF3855, RRID: AB_2836212; Affbiotech, Cincinnati, OH, USA) or anti-β-actin polyclonal antibody (1:2,000 dilution; bs-0061R; Bioss, Beijing, China). The PVDF membranes were subsequently incubated by horseradish peroxidase (HRP)—conjugated secondary antibody (1:2,000 dilution; bs-0295G-HRP; Bioss, Beijing, China). The band were visualized by chemiluminescence method using enhanced chemiluminescence (ECL) kit (NCM Biotech, Suzhou, China), and band intensity was quantified by AlphaEaseFC software. The relative expression levels of BCAS2 protein were calculated by average bands intensity relative to that of β-actin bands.

Immunohistochemistry

Testicular tissue samples fixed with 4% paraformaldehyde for 48 h were pruned into about 0.4 × 0.3 cm tissue blocks. After gradient ethanol dehydration, xylene transparency and paraffin wax dipping, the tissue samples were embedded to make paraffin wax blocks. After the wax blocks were trimmed, they were fixed on the clamping table of the microtome for continuous slicing (thickness 5 μm). Before affixed on slides, the cut slices were spread in warm water, followed by drying at 37 °C for 2 h. Immunohistochemistry for BCAS2 protein was carried out by a HistostainTM-Plus kit (Bioss, Beijing, China). Paraffin section was dewaxed by using xylene, dehydrated by gradient ethanol. The antigens were repaired by placed in 0.01 mol/L sodium citrate buffer solution, and the endogenous peroxidase activity was eliminated by adding dropwise 3% H2O2. After being blocked by using blocking solution, sections were incubated in a wet box with anti-BCAS2 antibody (1:500 dilution; Catalog No. DF3855, RRID: AB_2836212; Affbiotech, Cincinnati, OH, USA), and PBS replaced primary antibody as negative control. Subsequently, sections were incubated by secondary antibody as well as HRP-streptavidin, and positive signals (brown) of BCAS2 protein were visualized by a 3′-dia-minobenzidine (DAB) kit (Bioss, Beijing, China). After being counterstained by using hematoxylin and differentiated with 1% hydrochloric alcohol, sections were dehydrated with gradient ethanol and dewaxed by using xylene. Neutral gum was used to seal sections for further investigation and exploration. The representative images were captured by a biological microscope (Sunny EX31; Ningbo, China).

Immunofluorescence

Paraffin sections from testes of different development stages were prepared with the same procedure as immunohistochemistry. Sections were dewaxed and dehydrated by gradient ethanol, and then repaired by adding EDTA antigen repair buffer with microwave heating. After being blocked with 5% bovine serum albumin (BSA; Solarbio, Beijing, China), sections were incubated using anti-BCAS2 antibody (1:500 dilution; Catalog No. DF3855, RRID: AB_2836212; Affbiotech, Cincinnati, OH, USA) at 4 °C overnight. Subsequently, sections were incubated using fluorescein isothiocyanate (FITC)-labeled goat anti-rabbit IgG (1:200; Servicebio, Wuhan, China), and the nuclei were stained using 4′,6-diamidino-2-phenylindole (DAPI; Servicebio, Wuhan, China). Finally, sections were sealed utilizing solution containing anti-fluorescence quencher, and then observed and photographed under fluorescence microscope. Representative images were captured by CaseViewer software (3DHISTECH, Budapest, Hungary).

Statistical analysis

All experiment procedures were repeated independently at least three times. The relative expression levels of BCAS2 mRNA and protein were statistically analyzed by using one-way ANOVA in SPSS 21.0 software (SPSS Inc., Chicago, IL, USA). All data are presented in the form of bar charts as the mean ± SD. The p < 0.05 and p < 0.01 were regarded as statistically significant and extremely significant, respectively.

Results

Cloning and sequence analysis of Hezuo pig BCAS2 CDS

A specific target fragment of approximately 730 bp was acquired by the RT-PCR method using cDNA from Hezuo pig testes as templates (Fig. 1A). Sequencing results manifested that the cloned BCAS2 CDS sequence was consisted of 678 bp nucleotides translatable to 225 amino acids, with an ATG (M) start codon and a TGA stop codon (Fig. 1B). The resulting full-length CDS sequence of Hezuo pig BCAS2 has been deposited in GenBank (accession no. OP360012). One ORFs found in this cloned sequence using ORFfinder online tool in NCBI (Fig. 1C), and one conserved domain was discovered with the CDD online tool of NCBI, which was BCAS2 superfamily (amino acid residues located at 11–214, E-value of 3.42e−100; Fig. 1D). The CDS sequence alignment indicated that the cloned Hezuo pig BCAS2 sequence had three base substitution, namely, the replacement of A at nucleotide position 39 to G, the replacement of A at nucleotide position 288 to G, and the replacement of C at nucleotide position 447 to A, compared with the reference Sus scrofa BCAS2 sequence from the NCBI database (GenBank no. NM_001244424.1; Fig. 2).

Figure 1 Cloning and sequence analysis of Hezuo Pig BCAS2 CDS.

(A) PCR amplification product of BCAS2 CDS region. M, DL2000 maeker; 1 and 2, BCAS2 RT-PCR products. (B) The nucleotide and amino acid sequence of the cloned BCAS2 CDS region. (C) The analysis of open reading frame (ORF). (D) The prediction of conserved domains.

Figure 2 Sequence alignment between cloned and reference BCAS2 CDS region.

Molecular characterization of Hezuo pig BCAS2

As ananlyzed by ProtParam online server, molecular formula, total number of atoms, molecular weight, and theoretical isoelectric point (pI) of the protein encoded by Hezuo pig BCAS2 gene were C1143H1797N327O353S11, 3,631, 26,120.48 Da, and 5.48, respectively. The estimated half-life for the BCAS2 protein was 30 h in mammalian reticulocytes in vitro. The grand average of hydropathicity (GRAVY), extinction coefficients (γ = 280 nm), aliphatic index, and instability index (II) for BCAS2 protein were −0.688, 35,535, 77.64, and 44.68, respectively. The amino acid composition indicated that glutamic acid (11.56%) was the majority amino acid, followed by leucine (9.33%), alanine (8.89%) and glutamine (7.56%; Fig. 3A). The statistics revealed that hydrophilic amino acid residues (60.89%) were more than the hydrophobic amino acid residues (39.11%) in whole chain (Fig. 3B). At the nucleotide level, BCAS2 CDS sequence was composed of 33.30% base A (226), 18.70% base C (127), 24.80% base G (168), and 23.20% base T (157; Fig. 3C). The above results demonstrated that the BCAS2 protein of Hezuo pig was an acidic hydrophilic unstable protein. Netphos 3.1 Server analysis suggested that there were 19 potential phosphorylation sites, containing six threonine phosphorylation sites, four tyrosine phosphorylation sites, and nine serine phosphorylation sites (Fig. 3D).

Figure 3 Sequence composition and characterization of Hezuo pig BCAS2.

(A) Amino acid composition. (B) The number of hydrophobic and hydrophilic amino acid. (C) Base composition. (D) The prediction of phosphorylation sites.

The subcellular distribution of Hezuo pig BCAS2 protein was inferred by PSORT program, including 52.2% in cytoplasm, 26.1% in nucleus, 8.7% in endoplasmic reticulum, 8.7% in mitochondrial and 4.3% in vacuolar. The results indicated that BCAS2 was mainly distributed in the cytoplasm (Fig. 4A). The results of hydrophobicity analysis by ProtScale program in ExPasy server manifested that glycine at 9 position has the strongest hydrophobicity (score: 2.011), while the arginine at 41 position has the weakest hydrophobicity (score: −3.067). Therefore, this can be determined that the protein encoded by Hezuo pig BCAS2 gene is hydrophilic protein, which is consistent with the prediction results of ProtParam online server and statistical analysis (Fig. 4B). Hezuo pig BCAS2 protein had neither signal peptide sequence (Fig. S1) nor predicted transmembrane region (Fig. S2). The secondary structure predictions for Hezuo pig BCAS2 protein indicated that the protein had mixed secondary structures, consisting of 66.67% alpha helix, 30.22% random coil, 2.67% beta turn, and 0.44% extended strand (Fig. 4C). The tertiary structure analysis of coded protein revealed that it was mainly composed of alpha helix, random coil, beta turn and extended strand, which was similar with secondary structure (Fig. 4D).

Figure 4 Molecular characteristics and spatial structures of Hezuo Pig BCAS2.

(A) The subcellular localization. (B) The analysis of hydrophobicity. (C) Secondary molecular structure. (D) Tertiary molecular structure. The tertiary structure is based on the following models: Q9D287.1.A Pre-mRNA-splicing factor SPF27 (gene: BCAS2, organism: Mouse). Its global model quality estimate (GMQE) is 0.89, and sequence identity is 98.22%.

Homology analysis and evolutionary relationships among different species

The nucleotide and amino acid sequences of Hezuo pig BCAS2 gene were compared with other species, and as shown in Table S1, the homology of BCAS2 gene in mammals is above 87%. The nucleotide sequences of Hezuo pig BCAS2 gene have higher homology with Sus scrofa (99.56%), Arabian camel (95.28%), Bactrian camel (95.28%) and alpaca (94.99%), while the amino acid sequences have 100% similarity with Sus scrofa, Arabian camel, Bactrian camel, alpaca, sheep, goat, cattle, zebu cattle, water buffalo and Odocoileus virginianus texanus. The phylogenetic tree constructed by BCAS2 nucleotide sequences from Hezuo pig and other species displayed that Hezuo pig was the most closely related to Sus scrofa, followed by Arabian camel, Bactrian camel, and alpaca (Fig. 5).

Figure 5 Phylogenetic tree of the BCAS2 gene among different species.

The bootstrap values and branch lengths were exhibited above and below each branch, respectively. The closest homology with Hezuo pig BCAS2 gene is indicated by blue triangle.

Expression patterns of Hezuo pig BCAS2 at the mRNA and protein levels

The qRT-PCR results confirmed that Hezuo pig BCAS2 mRNA was universally expressed in different tissues including heart, liver, spleen, lung, kidney and testis, especially highly expressed in testis (p < 0.01; Fig. 6A). Its expression abundance in testis progressively increased with advancing age (Fig. 6B). A slight BCAS2 mRNA expression was detectable in the 30 d testes, whereas dramatically upregulated BCAS2 mRNA was found in the 120 d and 240 d testes (p < 0.01). At protein levels, BCAS2 expression profiles in developmental testes demonstrated a basically consistent trend as mRNA levels (Figs. 6C and 6D).

Figure 6 Expression patterns of Hezuo pig BCAS2 at the mRNA and protein levels.

(A) The relative expression levels of BCAS2 mRNA in various tissues at 240 d Hezuo pig. β-actin was used as a reference gene. (B) The relative expression levels of BCAS2 mRNA in Hezuo pig testes at different development stages. (C) Western blot bands of BCAS2 and β-actin in Hezuo pig testes at different development stages. β-actin was used as a loading control. (D) The relative expression levels of BCAS2 protein in Hezuo pig testes at different development stages. The bars represent means ± SD from three independent experiments. **p < 0.01, *p < 0.05, and ns, no significance. 30 d: 30 days old, 120 d: 120 days old, and 240 d: 240 days old.

Localization of BCAS2 protein in developmental Hezuo pig testes

To explore potential roles of BCAS2 during Hezuo pig spermatogenesis, positive BCAS2 protein signals were detected in developmental testes by using immunohistochemical staining. The results showed that BCAS2 protein was mainly observed in gonocytes for 30 d tesetes, while it was dominantly enriched in certain cells located in seminiferous tubules basement membrane for 120 and 240 d tesetes (Fig. 7). The basement membrane are lined with spermatogonia with round nuclei and Sertoli cells with irregular or triangle light-colored nuclei, which suggestive of that BCAS2 protein was predominantly present in spermatogonia and Sertoli cells for 120 and 240 d tesetes. Immunofluorescence staining was performed to verify the results of BCAS2 protein distribution and further determine its subcellular localization, and results indicated that BCAS2 protein was principally localized in nucleus of gonocytes, spermatogonia and Sertoli cells within seminiferous tubules (Fig. 8).

Figure 7 Immunohistochemical staining of BCAS2 protein in developmental Hezuo pig testes.

(A, D, G) Immunostaining patterns of BCAS2 protein (brown) in 30, 120 and 240 d Hezuo pig testes, respectively (200×); Scale bars, 100 μm. (B, E, H) Immunostaining patterns of BCAS2 protein (brown) in 30, 120 and 240 d Hezuo pig testes, respectively (400×); Scale bars, 50 μm. (C, F, I) Negative control with phosphate-buffered saline (PBS) replaced the primary antibody (400×); scale bars, 50 μm. The experiment was biologically repeated three times. 30 d: 30 days old, 120 d: 120 days old, and 240 d: 240 days old.

Figure 8 Immunofluorescence staining of BCAS2 protein in developmental Hezuo pig testes (200×).

(A, D, G) Nuclei were stained with 4, 6-diamidino-2-phenylindole (DAPI; blue); (B, E, F) BCAS2 protein were stained with fluorescein isothiocyanate (FITC; green); (C, F, I) merge. Scale bars, 100 μm. The experiment was biologically repeated three times. 30 d: 30 days old, 120 d: 120 days old, and 240 d: 240 days old.

Discussion

In this work, we firstly cloned the CDS region sequence of Hezuo pig BCAS2 gene. The results manifested that full-length CDS sequence of BCAS2 was consisted of 678 bp nucleotides, and encoded a total of 225 amino acids, which is consistent in length with initially study in human breast cancer cell lines (Nagasaki et al., 1999). To our knowledge, this is the first report regarding molecular characterization of BCAS2 in pig. A DNA sequence bounded by promoter and terminator is not necessarily true single gene product. In the absence of other information, DNA sequences can be read and translated by six different frameworks, which means that even when a gene’s DNA sequence is identified, it is often unclear what the corresponding protein sequence is (Sieber, Platzer & Schuster, 2018). The search for ORFs can reflect important regions of coding potential of a sequence. In this study, only one open reading frame was found in this CDS region sequence, indicating that this cloning sequence is the potential gene sequence encoding BCAS2 protein. This experiment found that Hezuo pig BCAS2 protein contained one conserved structural domains, BCAS2 superfamily. BCAS2 superfamily is comprised of some eukaryotic sequences of unknown functions, and mammalian members of this family are identified as BCAS2 protein. BCAS2 protein, as putative spliceosome associated protein, has capacity to regulate precursor mRNA splicing (Maass et al., 2002; Neubauer et al., 1998). In this study, it was found that although there were three base mutations in the BCAS2 nucleotide sequence of Hezuo pigs compared with the predicted sequence available from NCBI database, all of them were synonymous mutations and the translated protein was still invariable. Synonym mutations do not cause changes in corresponding amino acids, so most biologists have always regarded synonym mutations as neutral or nearly neutral, believing that they do not change the fitness of organisms including survival and reproductive ability, and will not be affected by natural selection (Kimura, 2007). However, a small number of existing studies have found that synonymous mutations can affect many biological processes except protein sequences, such as transcription factor recognition, mRNA splicing, folding and degradation, as well as the initiation, efficiency and accuracy of protein translation (Agashe et al., 2013; Shen et al., 2022). Therefore, whether these three point mutations affect the function of this gene in Hezuo pig testis remains to be further studied.

Among them, glutamic acid account for the largest proportion. Glutamic acid is an acidic amino acid (Wheeler & Wise, 1983), and its abundant presence further proves that the protein encoded by this sequence is acidic protein. What’s more, glutamic acid has a strong ability to form alpha helix (Marqusee & Baldwin, 1987), which is consistent with the prediction in this study that the secondary and tertiary structures of this sequence are mainly alpha helix by using SOPMA and SWISS-MODEL. The alpha helix proportion was as high as 66.67% in the Hezuo pig BCAS2 protein structure. The structure of alpha helix is rigid and principally hinges on hydrogen bonds to maintain stability, which supports overall conformation of some proteins (Makhatadze, 2005). Protein phosphorylation site is specific amino acid residue site where protein is phosphorylated. Most phosphorylated proteins encompass more than one phosphorylation site, which serine, threonine, and tyrosine are universal three sites in eukaryotes. Phosphorylation sites are critical for transport and function of protein, and phosphorylation at any site on protein can alter function and localization of the protein (Cao, Deterding & Blackshear, 2007). Although a total of 19 potential phosphorylation sites were predicted in this study, the specific phosphorylation sites still need to be further identified by western blotting and mass spectrometry (Bonilla et al., 2008). Additionally, the BCAS2 protein of Hezuo pig was found to be hydrophilic, unstable, non-transmembrane, and nonsecretory. The above molecular characteristics may be associated with complex function of BCAS2 (Idriss & Naismith, 2000; Hodge, Benhaim & Lee, 2020). Furthermore, this CDS sequence demonstrated a high homology (no less than 87%) compared with other published mammalian BCAS2 sequences from NCBI. The findings indicate that BCAS2 is highly sequence homology and evolutionarily conserved, which futher hint that the function of BCAS2 gene may be conserved in diverse mammals.

Among various regulatory factors in regard to testicular development and spermatogenesis, the specific expressions of genes in testes have an important role (Yang et al., 2021; Zhang et al., 2021). Consistently, the functional study on a gene during testicular development and spermatogenesis generally begins with expression research of the gene as well as its encoded protein in testis (Petit et al., 2015; Pinheiro et al., 2012). Hence, understanding the spatial–temporal expression patterns of BCAS2 gene at both transcriptional and translational levels contributes to unscramble its functions during testicular development and spermatogenesis in Hezuo pig. Herein, we investigated the expression patterns of BCAS2 mRNA in different tissues of Hezuo boars by qRT-PCR. Results showed that the transcript abundance for BCAS2 was universally discovered in multiple tissues including heart, liver, spleen, lung, kidney, and testis, which is consistent with the results in formerly published studies document that BCAS2 is widely expressed in various tissues, suggestive of its intricate role in multiple biological processes in organisms (Yu et al., 2019; Liao et al., 2015; Liu et al., 2017). The temporal expression results of BCAS2 transcript and protein in this study demonstrated that BCAS2 expression in Hezuo pig testes was gradually up-regulated with pig age, indicating its important role in testicular development and spermatogenesis of Hezuo pig. The expression trend of BCAS2 protein in immunohistochemistry and immunofluorescence is consistent with the trend of western blot results, both of which are increasing with ages. This change could not be observed at first glance on immunohistochemistry and immunofluorescence figures, which may be due to the inconsistent diameter of the seminiferous tubule in the testis at different stages, the inconsistent number of the seminiferous tubule presented at different photo multiples of immunohistochemistry and immunofluorescence, and the use of different batches of antibodies during the experiment. Based on these findings, we speculated that BCAS2 may exert important roles in Hezuo pig testis, and the variable temporal expression patterns of BCAS2 may be strong associated with its functional divergences during testicular development and spermatogenesis.

Substantial evidence suggests that BCAS2 plays a critical role in multiple biological processes. BCAS2 was first identified as a highly expressed oncogenic gene in breast tumors (Nagasaki et al., 1999; Maass et al., 2002), and was later found to be the core component of CDC5L/Prp19 splicing complex (Grote et al., 2010), which mainly takes part in pre-mRNA splicing (Chan & Cheng, 2005) and DNA damage response (Xu et al., 2015). It is closely related to cell survival and development (Chen et al., 2013; Chou et al., 2015), spindle assembly and the correct arrangement of chromosomes (Hofmann et al., 2013), and also participates in the development of early embryos (Xu et al., 2015). Recent studies in mice have shown that BCAS2 in germ cells was regarding alternative splicing in spermatogonia, and its disruption in mice will impair spermatogenesis, making it difficult to observe spermatocytes and meiosis events (recombination and synapsis), and eventually lead to male sterility (Liu et al., 2017). Protein, as product coded by genes, is carrier of biological activities as well as direct executors of biological functions, and its different distribution in tissues may be attributed to its diverse biological roles. Hence, in order to further investigate the potential roles for BCAS2 during testicular development and spermatogenesis, we detected the patterns of cellular distribution of BCAS2 protein in developmental Hezuo pig testes. As was suggested by immunohistochemistry and immunofluorescence analyses, the strong positive signal for the BCAS2 protein was observed to be mainly present in nucleus of spermatogonia at 120 and 240 d Hezuo pig testes. The similar finding has also reported in previous research, which documented that the intense BCAS2 protein is predominantly located in nucleus of spermatogonia in mice testes (Liu et al., 2017). In mice, BCAS2 gene modulates alternative splicing in spermatogonia especially splicing in the mitosis-to-meiosis transition during spermatogenesis, and its specific deletion in male germ cells affects splicing of several hundred genes regarding RNA processing, chromosome organization, regulation of transcription as well as sexual reproduction (Liu et al., 2017). The findings are indicative of a possible role for the BCAS2 gene in pre-mRNA splicing of spermatogonia during Hezuo pig testicular development and spermatogenesis. Nevertheless, in terms of molecular characteristics, the subcellular localization of BCAS2 predicted by PSORT program mainly existed in cytoplasm (52.2%), followed by nucleus (26.1%), which was inconsistent with the results of immunohistochemistry and immunofluorescence. It was speculated that on the one hand, the subcellular localization results obtained by computer were biased. Another possibility is that because BCAS2 is a gene that is ubiquitous in various tissues, small blood vessels or other components in the testes increase PSORT’s prediction.

Meanwhile, BCAS2 protein is also distributed in nucleus of Sertoli cells for 120 and 240 d Hezuo pig testes, in addition to its role as a spermatogonia regulator. Sertoli cell, as the only somatic cells in seminiferous epithelium, plays pivotal role in the unobstructed progression of germ cells to spermatozoa by supporting, protecting and nourishing spermatogenic cells as well as maintaining microenvironment for growth and development of germ cells (Griswold, 1998, 2018). Accumulating evidence indicates that genes localized in Sertoli cells play protective roles in proliferation and development of Sertoli cells (Claus et al., 2007; Wang et al., 2020; Park et al., 2021), suggesting BCAS2 may play a regulatory role in functional maintenance of Sertoli cells during Hezuo pig testicular development and spermatogenesis. In addition, BCAS2 protein signal was observed in nucleus of gonocytes from 30 d Hezuo pig testes. In postnatal animals, gonocytes, which represent germ cells in fetal and neonatal stages preceding formation of spermatogenic cells, migrate from central region to testicular basement membrane to relocate and differentiate (Culty, 2009, 2013). Current report demonstrated that genes distributed in gonocytes are associated with migration of gonocytes toward basement membrane (Basciani et al., 2008), morphological changes, maturation and survival of gonocytes (Ma et al., 2020; Xia et al., 2021) as well as gonocytes-to-spermatogonia transition (Pui & Saga, 2018). As we know from the above, BCAS2 might also be shown to be implicated in the proliferation or differentiation of gonocytes during Hezuo pig testicular development and spermatogenesis. Therefore, it is speculated that BCAS2 gene in testes of Hezuo pig may affect testicular development and spermatogenesis through its involvement in the proliferation or differentiation of gonocytes, pre-mRNA splicing of spermatogonia and functional maintenance of Sertoli cells. Although the specific mechanism of BCAS2 regulating testis development and spermatogenesis in Hezuo pigs still requires further exploration in cell and animal experiments using genome-editing technology, in this study, basic experiments such as cloning, qRT-PCR, WB, immunohistochemistry and immunofluorescence were conducted. The molecular characteristics, expression pattern and localization of BCAS2 gene in testis of Hezuo pigs were elucidated for the first time, and the role of BCAS2 gene in testis development and spermatogenesis of Hezuo pigs was preliminarily concluded, which filled the gap in the exploration of BCAS2 gene in Hezuo pigs. Additionally, the study of the BCAS2 gene in Hezuo pigs also provides a new avenue for veterinary scientists to further understand diseases or cancers related to reproductive disorders. The expression and regulation of genes related to reproduction can affect the development of testis and spermatogenesis (Bai et al., 2017; Yang et al., 2018), and the deletion of a certain gene often causes spermatogenic arrest. For example, Mice lacking the DAZL (deleted in azoospermia-like) gene are sterile and lack spermatozoa production, with only spermatogonia and a few spermatocytes (Schrans-Stassen et al., 2001). Spata2 (Spermatogenesis-associated protein 2) gene is one of the fundamental genes in the cellular signaling network that controls necroptosis and apoptosis, and its deletion will lead to increased expression of inhibin α and attenuated fertility in male mice (Masola et al., 2022). Combined with previous studies, these findings demonstrate that BCAS2 gene has potential physiopathologic and therapeutic implications in diseases or cancers related to male reproductive disorders.

Conclusions

In conclusion, this is the first report describing molecular characteristics of the Hezuo pig BCAS2 CDS region, along with first investigating its expression patterns and cellular localization at three different developmental stages. The full-length CDS region of the Hezuo pig BCAS2 gene was 678 bp in length and encoded 225 amino acid residues with highly sequence homology and evolutionary conservation to those of other mammalian species. The expression patterns of BCAS2 at mRNA and coding protein in Hezuo pig testes exhibited an age-dependent augmented expression pattern (p < 0.01). Moreover, BCAS2 protein was mainly present in nucleus of gonocytes at 30 d testes as well as spermatogonia and Sertoli cells at 120 and 240 d testes. On the basis of these findings, we concluded that BCAS2 may play pivotal roles in testicular development and spermatogenesis of Hezuo pig through its involvement in proliferation or differentiation of gonocytes, pre-mRNA splicing of spermatogonia and functional maintenance of Sertoli cells. However, the specific molecular mechanisms of BCAS2 gene during Hezuo pig spermatogenesis remain to be further investigated.

Supplemental Information

Supplemental Information 1 Author checklist.

Click here for additional data file.

Supplemental Information 2 Sequence data.

Click here for additional data file.

Supplemental Information 3 BCAS2 qRT-PCR raw data.

Click here for additional data file.

Supplemental Information 4 Full length uncropped gels.

Click here for additional data file.

Supplemental Information 5 The prediction of signal peptide of Hezuo pig BCAS2 protein.

Click here for additional data file.

Supplemental Information 6 The prediction of transmembrane domain of Hezuo pig BCAS2 protein.

Click here for additional data file.

Supplemental Information 7 Alignment of the similarity of nucleotide and amino acid sequence of BCAS2 CDS region between Hezuo Pig and other species.

Click here for additional data file.

Additional Information and Declarations

Competing Interests

Author Contributions

Data Availability

The authors declare that they have no competing interests.

Yuran Tang conceived and designed the experiments, performed the experiments, analyzed the data, prepared figures and/or tables, authored or reviewed drafts of the article, and approved the final draft.

Bo Zhang performed the experiments, prepared figures and/or tables, resources, and approved the final draft.

Haixia Shi performed the experiments, prepared figures and/or tables, and approved the final draft.

Zunqiang Yan conceived and designed the experiments, authored or reviewed drafts of the article, project administration, and approved the final draft.

Pengfei Wang conceived and designed the experiments, authored or reviewed drafts of the article, and approved the final draft.

Qiaoli Yang analyzed the data, prepared figures and/or tables, and approved the final draft.

Xiaoyu Huang analyzed the data, prepared figures and/or tables, and approved the final draft.

Shuangbao Gun conceived and designed the experiments, authored or reviewed drafts of the article, and approved the final draft.

The following information was supplied regarding data availability:

The sequence is available at GenBank: OP360012.

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
