# Peer review of "Molecular characterization, expression patterns and cellular localization of BCAS2 gene in male Hezuo pig"

_PeerJ, doi:10.7717/peerj.16341_

## Round 0.1 · original submission · Major Revisions

Dear authors. Thank you for your submission. At this time, I believe that this article requires extensive revisions in order to be published. Please, refer to the reviewers' comments for further details.

Reviewer 1 ·

Basic reporting

The English language could be improved to ensure that an international audience can clearly understand. In Figure 6A the word "kidney" is spelled incorrectly.

Experimental design

Investigating testes development of the Hezuo pig is interesting and so is the computational analysis of the protein and RNA sequence of BCAS2
However, the investigation of the expression pattern in situ is restricted to protein expression. The protein expression is shown in Figure 7 via immunohistochemistry and in Figure 8 via immunofluorescence which is redundant. Also, the WesternBlot in Figure 6 C suggests an increase of protein expression over time, which is not apparent from Figure 7 and 8.

Validity of the findings

The conclusion that BCAS2 plays a pivotal role in the testicular development in Hezuo pig can not be made based on the data presented. It is assumed that it has a similar role here based on findings from other model organisms.

Additional comments

The authors amplify the cDNA sequence of the gene, yet they do not investigate the RNA localization via RNA in situ hybridization. Relative mRNA expression via qRT-PCR provides information regarding the level of expression in a particular sample. However, it is not possible to draw conclusions regarding the pattern from the qRT-PCR when the entire organ and not individual cell types are used for the analysis.

Reviewer 2 ·

Basic reporting

The study outlines the importance of less reported BCAS2 gene in pigs. The authors finding on the this information could be useful for veterinary pathologists. The introduction and materials section is easy to understand. The authors have shared all the data required for the reviewers to validate their arguments.

Experimental design

The research largely falls under the scope of the journal. The research question and how this finding helps to fill the gap in research is under-discussed. I would suggest the authors improve the writing to largely focus on how this research is useful. The importance of the BCAS2 gene opens a new pathway for veterinary scientists to understand cancers since animal husbandry of pigs (since it concerns spermatogensis) is practiced across the globe.
Material and methods are sufficient to have a clear idea of what was done.

Validity of the findings

I have mentioned the details in the previous section.
The results are conclusive, but the authors should emphasize the importance of the research question rather than just report the findings. Since this forms the first line of the report on pigs, the overall language has to be improved to reach the audience.

Additional comments

1. Improvement in language and punctuation should be improved. I have highlighted some of them in my annotated pdf. Also, the importance of why pigs should be emphasized.
2. Line 74/75. "resulted in accumulation of pre-mRNA". Does this mean there is an accumulation of pre-mRNA? do they mean pre-slicing or does the mRNA accumulates without translation?
3. Line 92. "cancer malignant progression". Is it metastasis? What does it mean? cancer malignancy or cancer progression to the next stage? This is one example of why the language has to be improved.
4. Line 97. "there are few reports regarding BCAS2" - what do they say? This information is not highlighted in their introduction.
5. The authors have been using "pigs" and "boars" interchangeably throughout the manuscript. I am not sure Huzeo pigs can be called as boars. If not, change the terminology or sticky to a single term.
6. Line 204. Please describe how the paraffin sections were made. Did they use a microtome? What are the thickness of the tissue sections?
7. Line 222. Anti-BCAS2 antibody. What is the species specificity of the antibody they used?
8. Line 245-247. Authors have sequenced the BCAS2 gene from five different animals. Did they see any SNPs? OR they just sequenced the coding regions only? Please comment on the introns/exons of the BCAS2 gene (since the gene has both introns and exons and their title includes gene characterization)
9. The authors have looked at only the phosphorylation sites. Include glycosylation as they are important for several cellular processes like localization.
10. Line 276-279. With the development in protein structure with Alphafold, the authors should have used the Alphafold prediction instead of sticking to the classical structure prediction. And, how are those predicted structures were validated?
11. Line 283. Bootstrap values are pretty low in the phylogenetic tree. Authors should have tried tree construction using ML/MP methods. Also, the tree branching is deviating from tree of life (human and amphibian sequences are closely related). What is its significance?
12. Line 295. Why is there a correlation with the age of the testes? Is there any physiological significance?
13. Since p53 and BCAS2 exhibit direct interaction, reporting their co-localization would increase the impact of the manuscript. And authors should consider this.
14. Figures overall. Color schemes (especially with the localization) should be complemented with respected to the colorblindness chart. Color-blinded people could not differentiate between red and green. And, it is hard for me to observe the localization in Figure 8.
16. Elaborate if there is a literature report on dna-binding to BCAS2? Since BCAS2 is negatively charged protein, what is it doing in the nucleus? The importance of this observation is should be discussed.
17. Authors performed the qualitative analysis of IHC expression. The scoring or quantitative analysis of the IHC expression must be included and it has to be performed by a certified veterinary pathologist. Authors should make sure that the animal handling are done within the safety guidelines. I guess this is not mentioned in the methods section.
18. Again, nuclear localization of BCAS2 is not clearly identifiable from the color scheme mentioned in the figure.
19. I am just curious to know what are the closely related proteins to BCAS2? Any isoforms? What are its interaction partners?
20. Does it have any role in development? Will the KO of BCAS2 survive?
21. Any reports are available on the promoter and transcriptional regulation?
22. Fig. 7- what is the negative control used in this?
Please refer to the annotated copy for additional details.

Reviewer 3 ·

Basic reporting

Please see pdf attachment

Experimental design

Please see pdf attachment

Validity of the findings

Please see pdf attachment

Additional comments

Please see pdf attachment

Annotated reviews are not available for download in order to protect the identity of reviewers who chose to remain anonymous.

·

Basic reporting

This manuscript is clearly expressed, the background is detailed, and the structure of the manuscript is reasonable.But the references are cited a bit too much.

Experimental design

The details of the method need to be improved.

Validity of the findings

This study provides all the basic data and is statistically reasonable. The conclusions are clearly stated and relevant to the original research questions, but are not sufficiently refined, while some conclusions are inferred.

Additional comments

In this study, the authors cloned the porcine BCAS2 gene and identified the physicochemical characteristics and structure of BCAS2 protein by bioinformatics. The authors also analyzed the temporal differences in protein expression in the testis. In addition, by analyzing the cell distribution results combined with previous studies, it was inferred that porcine BCAS2 plays a key role in testicular development and spermatogenesis. The manuscript is well written and the discussion include the most relevant literature. Before acceptance some points need clarification.

1. Line 87: Since only one reference is cited in this sentence, it is not appropriate to use "several studies".
2. The authors should also briefly explain in the introduction why the BCAS2 gene should be studied in Hezuo pig.
3. Line 121: How are testes collected?
4. Lines 124-126: Why were the 240d pigs selected for tissue expression analysis?
5. Lines 143-144 and 182: The concentration of cDNA and primers needs to be specified.
6. Line 191: The amount of protein needs to be stated.
7. In Fig. 1A, the target fragment seems to be larger than 750bp.
8. Lines 277-279: The tertiary structure analysis needs to add what model it is based on, and what is the coverage and consistency with the model, respectively.
9. Lines 363-365: This sentence needs to be supported by literature.
10. Lines 418-421: Do all genes in gonocytes have these functions?
11. It is predicted by PSORT program that BCAS2 protein is highly likely to be distributed in the cytoplasm, which is not consistent with the results of immunohistochemistry and immunofluorescence in this study. Please briefly explain the possible reasons in the discussion.
12. There is too much scientific content in the discussion section, which needs to be reduced appropriately.
13. The title of figure 3 is not accurate because it involves nucleic acid sequence and amino acid sequence.

---

## Round 0.2 · Major Revisions

Dear authors, thank you for your refined resubmission. However, I do agree with one of the reviewers that some blocks of the manuscript should be revised for readability and clarity. Additionally, methodology should be as clear and complete as possible ( which include material and methods, incl antibodies cat #). And if protein expression visualized with IF and WB does not match, things should be re-checked. It may be due to different antibodies or even different lots, however this may be a case where you need to consider data reproducibility... If you are confident in the results already presented still please address the reviewers concerns and include such information in you discussion.

**Language Note:** The Academic Editor has identified that the English language must be improved. PeerJ can provide language editing services - please contact us at [email protected] for pricing (be sure to provide your manuscript number and title). Alternatively, you should make your own arrangements to improve the language quality and provide details in your response letter. – PeerJ Staff

Reviewer 1 ·

Basic reporting

Improvements to the language and grammar have been made, however, there are still a number of expression errors.
For example, line # 216 in the revised manuscript does not make any sense. Roasting is only used in the context of cooking. What is meant by the "before paraffin sections are made"?

Experimental design

The authors highlighted that IF was performed to better highlight the cellular localization of the protein. However, this is not apparent from the images presented.

Also, how have the different cell types been identified (gonocytes, spermatogonia, and Sertoli cells)?

The protein expression in both IHC and IF does not reflect what is seen in the Western Blot analysis. Given the level of expression and the number of cells shown at 30d in IHC, I would expect a clearer band and not just a smear in the western blot. Only one of the western blot gels shows that.

Validity of the findings

The molecular characterization of BCAS2 is interesting.
However, the expression pattern and cellular localization are not very clear from the data presented.

Reviewer 3 ·

Basic reporting

The authors have improved their manuscript a lot and I believe it should be published.

Experimental design

The experimental design is good.

Validity of the findings

The findings are of sufficient novelty for publication.

Additional comments

None

·

Basic reporting

No comment

Experimental design

No comment

Validity of the findings

No comment

Additional comments

Many thanks for the opportunity to read the revised version. All of my comments have been addressed and I have no further suggestions.

---

## Round 0.3 · accepted · Accept

Dear authors, I am happy to let you know that your manuscript is now acceptable for publication in PeerJ. Many thanks for your submission and diligent work.

Reviewer 1 ·

Basic reporting

N/A

Experimental design

N/A

Validity of the findings

N/A

Additional comments

N/A